# Adding Nano-TiO_2_ to Water and Paraffin to Enhance Total Efficiency of a Photovoltaic Thermal PV/T System Subjected to Harsh Weathers

**DOI:** 10.3390/nano12132266

**Published:** 2022-06-30

**Authors:** Miqdam T. Chaichan, Hussein A. Kazem, Ahmed A. Alamiery, Wan Nor Roslam Wan Isahak, Abdul Amir H. Kadhum, Mohd S. Takriff

**Affiliations:** 1Energy and Renewable Energies Technology Center, University of Technology-Iraq, Baghdad 10001, Iraq; miqdam.t.chaichan@uotechnology.edu.iq; 2Faculty of Engineering, Sohar University, P.O. Box 44, Sohar 311, Oman; h.kazem@su.edu.om; 3Department of Chemical and Process Engineering, Faculty of Engineering and Built Environment, Universiti Kebangsaan Malaysia (UKM), Bangi 43600, Malaysia; wannorroslam@ukm.edu.my (W.N.R.W.I.); sobritakriff@ukm.edu.my (M.S.T.); 4Faculty of Medicine, University of Al-Ameed, Karbala 56001, Iraq; amir1719@gmail.com; 5Chemical and Water Desalination Engineering Program, Department of Mechanical & Nuclear Engineering, Collage of Engineering, University of Sharjah, Sharjah P.O. Box 27272, United Arab Emirates

**Keywords:** additives, paraffin, PV/T system, nano-fluid, nano-paraffin

## Abstract

Iraq is characterized by hot and sunny weather with high radiation intensity. These conditions are suitable to produce photovoltaic electricity, on the one hand, but on the other hand are not suitable for photovoltaic modules whose efficiency decreases with increasing temperature. In this study, a photovoltaic module was practically cooled by two PV/T systems, one cooled by water and the other by nanofluid and nano-paraffin. Iraqi-produced paraffin was used in this study for its cheap price, and because its melting and freezing temperature (46 °C) is close to the operating range of photovoltaic modules. Nano-TiO_2_ was adopted as an additive to water and paraffin. The study results showed an obvious enhancement of the thermal conductivity of both water and paraffin, by up to 126.6% and 170%, respectively, after adding a 2% mass fraction of nano-TiO_2_. The practical experiments were carried out outdoors in the city of Baghdad, Iraq. A fluid mass flow rate of 0.15 kg/s was selected for practical reasons, since at this rate the system operates without vibration. The PV panel’s temperature, in the PV/T system (nano-fluid and nano-paraffin), decreased by an average of 19 °C when the tested systems operated during the peak period (12 PM to 3 PM). The decrease in temperatures of the PV module caused a clear improvement in its electrical efficiency, as it was 106.5% and 57.7% higher than the PV module (standalone) and water-cooled PV system, respectively. The thermal efficiency of this system was 43.7% higher than the case of the water-cooled PV/T system. The proposed system (nano-fluid and nano-paraffin) provides a greater possibility of controlling the heat capacity and increasing both efficiencies (electrical and thermal), when compared to a standalone PV module, in harsh Iraqi weather.

## 1. Introduction

The typical Iraqi citizen suffers from poor services in general and a severe shortage of electricity supply. Starting from 2003 till today, Iraqi citizens often use diesel generators extensively to fill the severe shortage of supplied electricity, especially after 2003. This focus on using such generators, without limits on the pollutants emitted by them, caused high levels of air pollution and noise rates [1]. Meanwhile, the sharp rise in the use of vehicles and trucks, due to the failure of the public transportation system, led to a doubling of the concentrations of many pollutants emitted from car exhaust [2]. The Iraqi government tends to, as a primary solution to the problem of air pollution, rely on building renewable electricity generation plants using photovoltaic cells. The Iraqi government is considering deploying 17 solar-powered power plants in different parts of the country, as shown in Table 1 [3]. However, the climatic conditions of this country, namely a significant rise in suspended dust in the air, as well as the continuous recurrence of dust storms, limit this trend towards the use of PV modules technology. Many research studies have put forward several suggestions to solve these problems. Many studies have emphasized the periodic cleaning of PV modules with various types of cleaning materials, according to the region and the human activities within it [4,5,6].

One of the reasons for the delay in installing these systems is that the performance of the photovoltaic modules deteriorates when the PV module temperature increases. The rise in PV panel temperature is largely from the solar radiation falling on it that is converted into heat, increasing the panel temperature, while a small part of this radiation is used to generate electricity [7]. Therefore, lowering the temperature of the PV modules enables them to operate close to their optimum performance. The researchers studied several methods of cooling PV modules such as air [8], water [9], nanofluids [10], PCMs [11], and nano-PCM cooling [12]. In the past decade, most researchers have emphasized the use of PV/T systems, which combine a photovoltaic module and a thermal collector. In this system, a heat exchanger (thermal collector) absorbs the excess heat in the photovoltaic body and delivers it to another application using a cooling fluid [13].

The literature is littered with several designs that have been suggested for cooling PV modules to improve their performance. Reference [14] used water to cool the PV/T system and improve its electrical efficiency. The authors confirmed a clear improvement in efficiency compared to the independent photovoltaic module. Reference [15] proposed the use of nanofluid, which is a base fluid, in water, with nanoparticles added to it, to enhance heat transfer rate from the PV module body to a thermal collector. This method has been studied in detail by many researchers, using many types of nanoparticles, with metallic and non-metallic sources, to discover the best fluids to use in PV/T systems for optimal performance [16,17,18]. Many studies have also focused on evaluating several types of base fluid, and examining their effects on the rate of heat transfer [19,20,21]. Most of the studies showed that water can be considered as one of the best of these basic fluids, because most of its thermal properties are suitable for PV/T applications [21].

Phase change materials store thermal energy when they undergo a change of state (phase) from solid to liquid, or vice versa. The heat stored, in this case (phase change), is latent heat, and this process is called the charging process, during which this material melts when its temperature reaches the melting point (the temperature of its phase change) [22]. The temperature of the substance remains constant until the end of the melting process. When the molten material is cooled, it begins to release the heat stored in it during the vacuum process [23]. Researchers have taken advantage of this property in many solar energy applications. In such applications, the temperature fluctuates due to the oscillation of the incident radiation from sunrise to sunset, and the interruption of heat when the sun is absent. Therefore, researchers have used PCMs in solar thermal applications to store heat in solar stills [12,24], air heaters [25,26], and PV/T systems [27,28]. Many phase change materials are available, each with different melting and hardening points, ranging from 5 °C to 190 °C. These materials store latent heat about 5 to 14 times as much as sensible heat storage materials, such as water, pebbles, or rocks [29].

Paraffin is one of the PCM families, and at room temperature takes the form of wax, which is formed of hydrocarbons produced in the distillation of crude oil. Paraffin wax is one of the most important types of PCM used in research for heat storage applications [30], because it is cheap and can store moderate heat energy, but it is criticized for its low thermal conductivity (TC) [30]. In general, paraffins are safe, affordable, non-irritating, and have a wide range of melting and solidification temperatures making their use possible in many latent heat storage applications. These materials are chemically inactive and stable. In addition, applications using paraffin for heat storage maintain high stability of melt-hardening cycles over very long periods [31]. As for some properties that may be considered undesirable, such as low TC, melting temperatures, and moderate flammability, they can be partially treated by adding additives such as nanoparticles or paraffinic materials.

Reference [32] used TRNSYS, with empirical confirmation of the extent of the decrease in the surface temperature of the PV panel, because of adding PCM to the PV/T system. The low TC of paraffin has been identified as one of the drawbacks that could limit their use in PV/T systems [33]. Therefore, several studies have suggested adding high TC nanoparticles to paraffin to improve the low TC of these materials [34,35].

Most nanoparticles, especially metallic ones, are characterized by their high TC [36,37,38]. Therefore, adding nanoparticles to the base fluid or paraffin will significantly improve their TC. However, these particles must be added in limited weight or volume ratios, as increasing their percentage in the basic fluid or paraffin will greatly reduce the stability of the nano-fluid or nano-paraffin [39]. Stability means the persistence of the thermophysical properties of a nanofluid or nano-paraffin for a long time. This is achieved by ensuring that nanoparticles do not collect and clump, and then precipitate for a period of time, instead remaining suspended in the base fluid or paraffin. The agglomeration of nanoparticles causes an increase in their mass, which causes their sedimentation, and, as a result, there is a clear decrease in the TC of the nanofluid or nano-paraffin [40]. The mixing process can be considered costly in terms of material, as well as in terms of the time it takes [41]. Ghademi et al. [42] considered that van der force is responsible for the agglomeration and deposition of these nanoparticles. Reference [33] confirmed that the deposition of nanoparticles causes the deterioration of the quality of the nanofluid. Therefore, ensuring the stability of nano-mixtures (whether nanofluids or nano-paraffin), and their uniform distribution within these mixtures, is a key factor in the quality of the nano-product.

The best way to mix nanoparticles with a base fluid or paraffin, which most researchers recommend, is to use ultrasonic vibration with the addition of a surfactant to the base fluid or molten paraffin, to achieve a product with good stability, i.e., stable for a long period of time [43]. Numerous experimental studies that used this method found it very effective, and the researchers were able to obtain nano-products with acceptable stability. Some researchers emphasized the sonication time and considered it a key factor in achieving a stable nano-product. It can be said that all studies did not agree on a specific time for sonication, and this is normal, as there are many differences, such as the base materials or the quality of the added nanoparticles, which could cause such differences. Afzal et al. [43] confirmed this matter, as they considered that the time of the sonication process is determined by the type of nanofluid, especially the added nanoparticles, i.e., the percentage, shape, and size of the particles. Chen et al. [44] mixed nanoparticles with water with several periods of sonication (2 h and 45 min, 3 h and 15 min, and 3 h and 45 min). The authors found that the optimal sonication time for the prepared liquids was 3 h and 15 min.

Thermophysical properties determine the rate of heat transfer by the nanofluid or nano-paraffin. These properties can be specified as TC, viscosity, density, and specific heat [45]. These properties change with the temperature of the nanofluid or nano-paraffin, which determines the efficiency of their performance in the heat transfer process [46,47,48]. In addition to the temperature of the nano-product, other factors determine the quality of its thermophysical properties (including, but not limited to, the type of base liquid, the type of nanoparticles used, the temperature range of application, the sonication period used to prepare the nanofluid, and the percentage of nanoparticles added to the base fluid) [49,50,51]. Researchers have not yet agreed on a clear definition of the effect of any of these factors on the thermophysical properties of nanofluid and nano-paraffin. Therefore, researchers continue to evaluate and study these factors to this day, aiming to determine the optimal nano-fluid and nano-paraffin for use in solar heat transfer applications.

In this empirical study, the use of PV/T systems in the harsh environmental conditions of Iraq for cooling PV modules was investigated. The PV/T system was designed in the form of a tank, filled with paraffin, attached to the PV panel at the back to absorb excess heat from it and reduce temperature fluctuation on its surface. Inside the wax, a heat exchanger was dipped, through which a nanofluid with high TC circulated, which in turn absorbed the heat collected in the paraffin and expelled it out of the system. The objective of this study was to improve the electrical efficiency of the PV/T system and make it suitable for working in harsh and hot weather conditions, such as those in Iraq.

## 2. Materials and Methods

Figure 1 details the practical steps involved in the study, starting from the selection of materials and preparation of nanofluids and nano-paraffin, to the tests that were carried out on them, as well as the measurements taken from the systems, which will be explained in detail in the following paragraphs.

### 2.1. PV/T System

In this study, a nano-paraffin tank was designed, fabricated, and attached to the PV panel’s (Table 2) backside. Inside this tank, a heat exchanger (Figure 2A) was placed to circulate the coolant, which absorbs the heat stored in the paraffin, perpetuating the heat transfer process from the PV to the paraffin (Figure 2B). This tank was connected to the photovoltaic module by welding. The space between the tank surface and the back of the module was filled with silicone oil, to facilitate the heat transfer process and ensure that no air gaps occurred that could prevent or reduce this transfer. The studied system consists of three photovoltaic modules; one of them is standalone, the second is a PV/T water-cooled system, and the third is heated with nano-paraffin and nano-fluids. The photovoltaic panels were all set facing south at an angle of 33°, to suit the Baghdad city location. The studied system also contains two water pumps to circulate the water and nanofluid, a container for the nanofluid, a data acquisition system, and a laptop computer. The paraffin container was insulated on all sides exposed to the air using insulation (glass wool, 2.5 cm thick) to conserve all heat energy absorbed from the PV panel. For consistency of results, accounting for the fact that the experiments were carried out outdoors, data collection, from measurements of all three systems, was carried out at the same time.

The performance of the treadmill systems was calculated using the following equations.

Collected thermal energy:$$Q_{u}=\dot{m}C_{p}\left(T_{o}-T_{i}\right)$$
where *Q_u_* is the useful thermal energy gained from the water or nanofluid, *C_p_* is the water or nanofluid specific heat (J/kg K), and *T_o_* and *T_i_* are the outlet and inlet temperatures, respectively.The PV/T system thermal efficiency:$$\eta_{th}=\frac{Q_{u}}{I_{s}\times A_{co}}$$
where $\eta_{th}$ is the thermal efficiency, *I_s_* is the solar radiation intensity, and *A_co_* is the collector surface area.Electrical power (W):P=I×V
where *P* is the electrical power, *I* is the electrical current, and *V* is the voltage.Electrical efficiency: $$\eta_{el}=\frac{P}{I_{s}\times A_{p}}$$
where $\eta_{el}$ is the electrical efficiency, *P* is the power, *I_s_* is the solar radiation intensity, and *A_p_* is the PV panel area.The total efficiency:$$\eta_{t}=\eta_{el}+\eta_{th}$$
where $\eta_{t}$ is the total efficiency.

### 2.2. Materials

The choice was made to use nano-TiO_2_, which has an acceptable TC; it was added to Iraqi paraffin, which is available in local markets, to form nano-paraffin, and to water, to form a nanofluid. The paraffin used is affordable (US$2/kg). Nano-TiO_2_ has important properties, such as chemical stability, high surface area, suitable electronic band structure, and high quantum efficiency. This type of nanoparticle has been used in applications such as dyes for photovoltaic cells, biomedical implants, and other applications [52,53]. Nano-TiO_2_ is available in local markets at a price of less than US$2/g [54,55]. Table 3 lists the properties of the nano-TiO_2_ used in this study. Table 4 illustrates the characteristics of the paraffin used in the practical tests.

The nanoparticles were prepared by placing them in an oven at a temperature of 220 °C for 15 min, to remove any moisture that might be present. This drying process is important in confirming that there is no interconnection between nanoparticles due to moisture before adding them to paraffin or water, in any proportion, even a very small amount. The same mass fraction of nanoparticles was added to water and paraffin (0.5%, 1.0%, 1.5%, and 2.0%). A sensitive balance (type EJ6I0-E) with a measurement accuracy of 0.0001 g was used.

### 2.3. Preparation of Nano-Paraffin

The paraffin was heated to 60 °C and placed in a container inside a bath that vibrated at ultrasonic speed, while keeping the paraffin hot throughout the mixing period, to eliminate the impact of its viscosity on nanoparticle diffusion. Nano-TiO_2_ particles were gradually added to the vibrating liquid paraffin while the sonication process continued, for two consecutive hours without stopping. In this study, an ultrasonic vibrator (type TELSONIC ULTRASONICS CT-I2) with a 12-L bath and an electrical heater with a capacity of 800 W was used. This vibrator could vibrate the fluid in the bath with a maximum frequency of 80 kHz. The sonication time was extended in the preparation of nano-paraffin to ensure a wider spread and optimal distribution of nanoparticles through the paraffin, which hinders future agglomeration. This process succeeded in several studies, and was praised and considered the best method for mixing nanoparticles with PCM [56].

It could be confirmed that the mixing process was successful by seeing the paraffin’s color completely changed and without impurities, indicating that the nanoparticles spread through it and were not deposited. Completely changing the color of paraffin without impurities is an indicator of good diffusion of nanoparticles in paraffin.

### 2.4. Preparation of Nanofluid

Several measures must be taken to ensure that a sufficiently stable nanofluid is prepared. The steps of mixing nano-TiO_2_ with water used by reference [51] were adopted. Nano-TiO_2_ was added to the water in the same ultrasonic shaker that was previously used to prepare nano-paraffin. The mixing process here lasted longer than the previous case, as the ultrasonic shaking continued for three and quarter and hour, to ensure the best diffusion of nanoparticles in the suspension [44]. The study also relied on adding 0.1 mL of surfactant (Cetyl Trichromyl Ammonium Bromide (CTAB)) to the water to ensure higher stability of the suspension. In this study, the results of reference [57] were used to inform the mixing method, the amount of surfactant added, and its type.

### 2.5. Uncertainty Analysis

In practical studies, uncertainty analysis is very important to ensure the validity of the measured data. Uncertainty is analyzed through practical calibration of all measuring devices, such as thermocouples, and measurement of thermophysical properties (viscosity, density, heat capacity, and TC). Instruments were calibrated and their results compared with a standard instrument; the deviation from the standard readings was considered an uncertainty [58]. Table 5 shows the details of the measurement tools used and the uncertainty for each. The study uncertainty was assessed by employing the Kline and McClintock equation [58]. The TC and heat capacity were measured using a KD2 Pro-Analyzer, while, for density measurements, a Density tester (type DII-300 L) was employed. A Brookfield programmable viscometer (model: LVDV-III) was used to measure both prepared materials viscosities, while a Zeta-Sizer Nano Analyzer (ZSN) was used to measure the prepared nanofluids’ Zeta potentials.

The results show that the uncertainty of the measurements of the current study was less than 5%, which means that the accuracy of the measurements made is geometrically acceptable. In addition, each experiment was repeated three times to ensure repeatability, and the arithmetic mean of the results was taken.
$$W_{R}={\left[{\left(\frac{\partial R}{\partial x_{1}}w_{1}\right)}^{2}+{\left(\frac{\partial R}{\partial x_{2}}w_{2}\right)}^{2}+\ldots +{\left(\frac{\partial R}{\partial x_{n}}w_{n}\right)}^{2}\right]}^{0.5}$$

Hence, the experiments in the study are provided as below:$$W_{R_{1}}={\left[{\left(1.16\right)}^{2}+{\left(0.72\right)}^{2}+{\left(0.8\right)}^{2}+{\left(1.04\right)}^{2}+{\left(1.3\right)}^{2}+{\left(0.94\right)}^{2}+{\left(0.55\right)}^{2}+{\left(0.88\right)}^{2}\right]}^{0.5}=3.42$$

## 3. Results and Discussion

Figure 3 shows photographs of samples of the used paraffin and nano-paraffins. The far left shows a sample image of the Iraqi paraffin used in the study, which has an average melting point of 46 °C. This paraffin is light brown, due to the low number of hydrocarbons in its chemical composition (it did not exceed 20 carbon atoms) and its low oil content. When adding nano-TiO_2_ in the smallest amount (0.5%), its color changes to white, and it becomes whiter with increasing mass fractions of added nanoparticles. If all paraffin turns white, this means complete and successful mixing; the presence of points or areas of light brown color means the mixing was a failure, and there is a need to repeat the sonication process.

Nano-paraffin and nanofluids thermophysical properties. Table 6 and Table 7 represent the measured thermophysical specifications of the prepared nanofluids and nano-paraffin. In the next section, these properties will be discussed in detail.

### 3.1. Thermal Conductivity

TC increased when the mass fraction of added nano-TiO_2_ was increased, in both water and paraffin. Titanium oxide is a highly conductive metal oxide, so adding its nanoparticles to water caused a clear enhancement of the nanofluid TC, amounting to 36.6%, 88.3%, 108.3%, and 126.6%, resulting from 0.5%, 1.0%, 1.5%, and 2% nano-TiO_2_ mass fractions added to water, respectively. As for paraffin, a quite remarkable increase in its TC at room temperature (25 °C) was measured. Of course, conductivity of the liquid state decreases by a certain percentage, but with nano-additives it remains high. Reference [59] found a decrease in the maximum value of TC when the material reached the phase change stat; after completing this change, the conductivity increased in both paraffin alone and with nano additives. Nano-TiO_2_ addition caused increments in the TC of nano-paraffin by 55%, 140%, 170%, and 230%, from 0.5%, 1.0%, 1.5%, and 2% nano-TiO_2_ mass fractions added to paraffin, respectively. These results indicate that the addition of nano-TiO_2_ to paraffin is more efficient than adding it to water, because of paraffin’s solid state at the measured temperature.

### 3.2. Density

The nanofluid and nano-paraffin density increased with the addition of nanoparticles. However, as the added nanoparticles mass fractions were small, the resulted density variations were also small. For the nanofluids, the density increments were 1.25%, 1.9%, 2.2%, and 2.5%, from 0.5%, 1.0%, 1.5%, and 2% nano-TiO_2_ mass fractions added to water, respectively. Likewise, for paraffin, the addition of 0.5%, 1.0%, 1.5%, and 2% nano-TiO_2_ mass fractions caused its density to increase by 1.2%, 1.88%, 2.12%, and 2.38%, respectively. 

### 3.3. Viscosity

Viscosity expresses the resistance of a fluid to flow when there is a pressure difference that forces it to move. In PV/T applications, when paraffin is used, there is no movement, so the effect of viscosity here is very limited. However, the state of paraffin changes from solid to liquid during the melting process (charging), and then from liquid to solid during the solidification process (discharge). Viscosity plays an important role during the liquid paraffin period, as nanoparticle dispersion depends on it. Therefore, this property was studied during the increase in temperature of the nano-paraffin, to ensure its behavior during this period. Table 7 shows the results of the change in viscosity of the heat storage materials (paraffin and nano-paraffin) when temperatures were raised from 25 °C to 65 °C. These temperatures were chosen because they are the temperatures at which PV/T systems operate in most locations. The results showed that the viscosities of nano-paraffin with variable mass fractions are close, because the mass fractions of nanoparticles added are very small. The addition of NanoTiO_2_ increased viscosity, whether liquid or solid, compared to paraffin. The viscosity of nano-paraffin decreased rapidly when the temperature increased (due to its TC amelioration compared to paraffin, as well as the improvement of thermal energy diffusion during heating). The decrease in density and viscosity with increasing temperatures (Table 7) means that water and nanofluid circulation pumps do not need to draw additional electrical power. This result is a positive one for any large system, as its generated power will not be affected because of circulating nanofluid with higher viscosity and density; these two characteristics will decrease with higher operating temperatures during the day.

### 3.4. Nanofluid Stability

In this study, a zeta potential analysis was adopted to measure the stability of the prepared suspensions, while another method was adopted to measure the stability of nano-paraffin, as will be explained in the next paragraph. In this technique, the change of electric charges in the nanofluid is measured, as the oppositely charged nanoparticles are attracted to the free charges. This process takes place in the nanofluid. The zeta potential values express the stability of the measured nanofluid. For example, a zeta value of more than 60 mV expresses very high stability of the nano-suspension. In the case of a zeta limit between 40 and 60 mV, the stability of the nanofluid performs well. However, if the zeta voltage drops to between 30 and 40 millivolts, the nano-suspension is considered to have acceptable stability. If the zeta potential reaches less than 30 mV, the nano-suspension is considered unstable. Table 6 shows the zeta potential measurement of the prepared nanofluids. All prepared suspensions had a small mass fraction, so their stability was high, and the most stable suspension was that with a mass fraction of (0.5%). The addition of nano-TiO_2_ particles with minute sizes (20 to 50 nm) and a high surface area, carefully dispersed using the sonication technique, which lasted for a sufficient time, caused uniform and fair distribution in the base liquid. The lowest zeta potential (48 mV) was observed in the case of adding 2% nano-TiO_2_ to water. This zeta potential indicates that the nanofluid has a very good stability.

### 3.5. Nano-Paraffin Stability

The stability of any nano-paraffin production presents a real challenge. Today, with the use of mixing by sonication, for sufficient duration, and the use of small-sized nanoparticles, this process has become easier, and nano-paraffin product have high stability. The processed nano-paraffin undergoes a large number of repeated heating and cooling processes for a long period of time. The instability of nano-paraffin causes the deterioration of its thermal conductivity with time, which negatively affects the performance of the PV/T system. The success of using nano-paraffin in PV/T systems depends entirely on enhancing the mixing of nano-particles with paraffin and the stability of the mixture for a long period of time, while reducing costs. In this study, the method used by reference [59], in which the thermal conductivity of nano-paraffin is measured at equal time intervals, was adopted to determine to what extent it can be effectively used. There is a high potential for the agglomeration and deposition of nanoparticles in a nano-paraffin tank, and the most appropriate course of treatment is the optimum distribution of the particles through all the paraffin in the tank. Due to limited study time, conductivity deterioration by 5% was adopted as an indicator of declining stability, with the possibility of utilizing this mixture (Nano-paraffin) until its conductivity decreased to 30%, after which it would be safer to re-mix again, so as not to affect the performance of the PV/T system.

The results listed in Table 7 show that the stability of nano-paraffin mixtures was very good, and exceeded 85 days before their conductivity decreased by 5%. This result indicates the possibility of using it for long periods before emptying the mixture from the tank and re-mixing it again. The highest stability was recorded for the 0.5% mass fraction-added mixture (98 days), and the lowest was recorded for the 2% mixture (88 days). From here, it is possible to confirm the success of the method used in this study for mixing nano-TiO_2_ with paraffin, as well as the good selection of materials and mixing time.

### 3.6. PV/T System Performance

Figure 4 shows the change in solar radiation intensity during the time of the experiments. The figure illustrates the data measured on site. It can be noticed from the figure that the solar radiation intensity in the city of Baghdad is very high, and its peak is between 12 PM and 2 PM, when it reaches about 950 W/m^2^. Solar radiation fluctuates as a result of environmental conditions, such as the movement of clouds and air masses, so the rise and fall of this intensity is not consistent. This high irradiance is preferred for the production of photovoltaic electricity (close to 1000 W/m^2^ under standard conditions). However, the ambient temperature is higher than standard conditions (25 °C), reaching about 40 °C. These conditions cause the temperature of the photovoltaic panels to rise, which leads to deterioration of the power levels generated by them.

The first practical experiments were conducted to define the most suitable flow rate of coolant (whether water or nanofluid) in the heat exchanger. Therefore, this flow was changed to several settings by adjusting the locking hole and measuring the fluid mass flow rate (Figure 5). The flow rates of water and nanofluids were measured using HC (US Hunter), with an uncertainty of ±0.55 (as illustrated in Table 5). The experiments were performed using water. Figure 4 demonstrates that the rising fluid flow rate caused a reduction in PV module body temperature. The increase in mass flow rate caused an increase in absorbed heat; however, as the vibration begins to occur in the system along with a large increase in flowing mass, there are limits to this increase. The results show that the best photovoltaic panel temperature reduction occurred when 0.175 kg/s (Re No. = 308,000) was used, but the vibration accompanying this flow caused it to work better at the lower flow rate of 0.15 kg/s (Re No. = 282,000), without vibration of the system. All of the following experiments took place at this flow rate.

Figure 6 shows the PV module surface temperature variations according to the change in the type of cooling used. The standalone PV module reached its highest possible temperatures (above 55 °C) at peak time, from 12 PM to 3 PM. When cooling this module with water, its temperatures dropped significantly at peak time, reaching 45 °C (10 °C less, on average). When using the nano-fluid and nano-paraffin PV/T system, the temperature decreased further, reaching 36 °C (an average of 19 °C less than the standalone PV panel). 

The proposed PV/T system (nano-paraffin and nano-fluid) has superior cooling capabilities than the water-cooled PV/T system alone. From the figure, it can be seen that, for the period before 9 AM, the nanofluid-nano-paraffin cooling is limited, and the module temperature is high. This phenomenon can be explained by the fact that the paraffin temperature, at this time, didn’t reach the melting point, and the differences between the PV surface and paraffin temperatures were small. However, despite this, the temperature of the PV/T module remained lower than the temperature of the standalone PV. 

Figure 7 shows the effect of the changing surface temperature of the PV module, due to weather conditions and exposure time, on the PV modules voltage. The shape of the voltage waves for both PV and PV/T systems is significantly increased by an increase in the intensity of the solar radiation, and this change is greater than the current condition, as mentioned by many studies [11,33,38]. The figure shows the presence of fluctuations in the voltage wave with time, as a result of the changing intensity of solar radiation, which is clearer in the case of the standalone PV module, while the PV/T system (nano-fluid and nano-paraffin) is flatter with the change of time. Any change, even a small one, in solar radiation intensity directly affects the generated electrical voltage. The measured voltages range from 10–11 V in the case of a standalone PV module, 12–13 V in the case of a water-cooled PV/T system, and 16–18 V in the case of nano-paraffin and nanofluid PV/T system.

Here it must be explicitly stated that the energy spent by the water and nanofluid circulating pumps was not included in the calculations of energy consumption of the photovoltaic modules studied. This behavior was due to the use of one panel as a representative of each studied system (the produced power of the panels is limited), which reduced the productivity of the panels and hindered fair comparison between them. In the case of larger systems, the consumption of the circulation pump must certainly be included in the calculations. 

Figure 8 shows the differences in electrical efficiency among the studied systems. The standalone PV module efficiency is highest in the early morning, then decreases, reaching its lowest values at peak time, but recovers some of its losses before sunset. The dependence of this module on air cooling and its movement is not guaranteed, especially in the city of Baghdad, as wind speed ranges from 0 m/s to a maximum of 3 m/s, which caused significant heating of this module and a clear reduction in electrical efficiency. As for the PV/T system (water-cooled), its electrical efficiency is higher than the previous case, with an average of 30.9% for a full-day operation. The curves show that the electrical efficiency of this system declined at peak time. As for the last case (the nano-fluid and nano-paraffin cooled PV/T system), the electrical efficiency is higher by an average of 106.5% and 57.7% compared to the two previous cases (the standalone PV and water-cooled PV/T systems), respectively. This result demonstrates the high efficiency of the cooling process in this system.

Figure 9 shows the differences in thermal efficiency among the studied systems. There is no thermal efficiency for the standalone PV module, as the heat absorbed from it is not utilized. As for the case of PV/T systems, this heat can be collected and used in many applications, the most important of which being heating water for domestic purposes, to heat swimming pools, or in solar stills. The water-cooled PV/T system efficiency is lowest in the early morning, reaches its maximum values at peak time, and decreases, relatively, after that until sunset. In the case of the nanofluid and nano-paraffin PV/T system, the thermal efficiency is little, and almost equal to the previous system, in the early morning hours, but it increases clearly and exceeds the previous system by 43.7% during peak period. This result supports our previously reached conclusion, which is that the cooling process of this system is very efficient.

Figure 10 shows that the nano-paraffin nanofluid PV/T system produced the highest total efficiency compared to the other two studied systems. This system, which had the highest cooling capacity for the photovoltaic panel, compared to the other systems (Figure 5), produced the highest thermal efficiency (Figure 8) and electrical efficiency (Figure 7), and, as a result, its total efficiency was the greatest. The total efficiency is the sum of the electrical and thermal efficiencies, so we find that it is the highest possible during the peak period (from 12–4 PM) for the two studied PV/T systems, while for the case of the PV system it is at its lowest values, because this system does not cool and does not have thermal efficiency. The highest total efficiency was 84.4%, for the nano-paraffin and nanofluid PV/T system, at 2 PM. When cooling with water, the maximum total efficiency was 57.1%, at the same hour. As for the highest total efficiency of the standalone PV system, it was 11% at eight o’clock in the morning, and this efficiency indicates an electrical efficiency. It is the highest possible at the coolest temperature of the PV panel, which is of course the beginning of exposure to solar radiation.

### 3.7. Comparison with Other Works from Literature

Figure 11 shows the ameliorations in the TC of different types of nanofluids for variables studies from the literature. The results show that nanofluids prepared from water + nano-TiO_2_ (recent study) provide the best TC enhancement rate. The reason for this (despite the presence of nanoparticles with a higher conductivity than nano-TiO_2_) is the use of particles of small size, which can stay suspended for as long as possible. In addition, the sonication process used was successful in terms of shaking speed and duration.

Figure 12 shows the improvement in the TC of several different types of nano-PCMs. The results show that the nano-paraffin prepared in this study provided a clear improvement in TC, and ranked second after the results of reference [64]. The successful mixing process and quality of the PCM used, in addition to the quality of the nanoparticles, are the reasons for obtaining a stable nano-PCM with a high TC. The results show the probability of enhancement using paraffin, among all types of PCMs used.

## 4. Conclusions

Most of the time, the weather in Iraq is dusty, hot, and highly radioactive. These matters limit the possibility of relying on PV power plants with high capacities. In this study, a practical attempt is made to reduce the PV module’s temperatures by cooling them in two different methods, the first using a cooling fluid (water), while in the second a system consisting of a thermal tank containing nano-paraffin and cooled with a nanofluid was employed. Nano-TiO_2_ was selected, for its particles of small size and diameter, to be added to water and paraffin. This addition significantly improved the TC of water (126% higher compared to water, when 2% Nano-TiO_2_ was added). In addition, the paraffin TC was improved by up to 170% at the same mass fraction of added nanoparticles. The results of the study show a clear decrease in the temperature of the photovoltaic panel of the proposed system compared to the water-cooled PV/T system or standalone PV system. During peak time (from 12 PM to 3 PM), the temperature of the PV panel decreased by about 19 °C. The proposed system produced the highest electrical efficiency, with an increase of 106% and 56% compared to the standalone PV panel and water-cooled system, respectively. The nano-system achieved 43% higher thermal efficiency than the water-cooled system.

## Figures and Tables

**Figure 1 nanomaterials-12-02266-f001:**
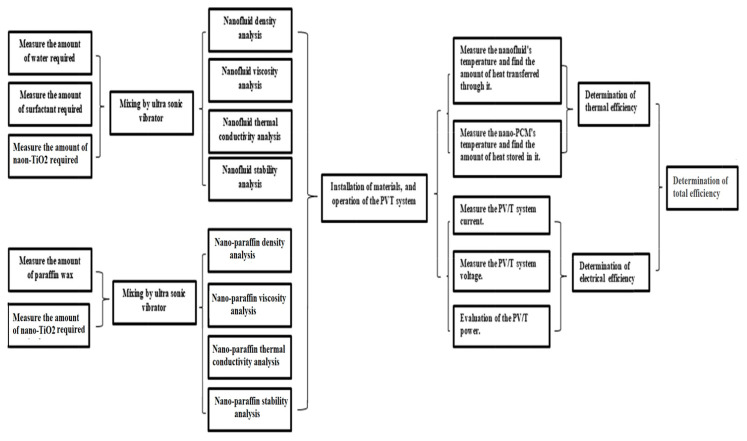
Block diagram for practical steps conducted in the study.

**Figure 2 nanomaterials-12-02266-f002:**
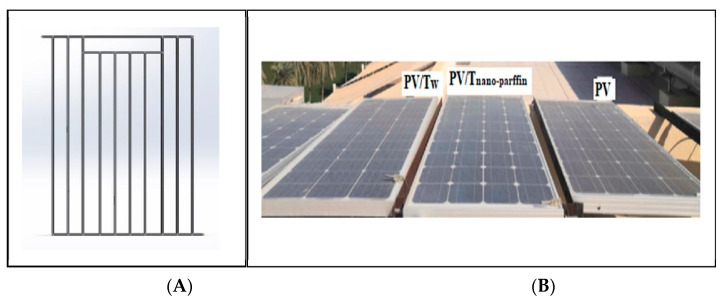
(**A**) nanofluid heat exchanger, and (**B**) the studied systems.

**Figure 3 nanomaterials-12-02266-f003:**
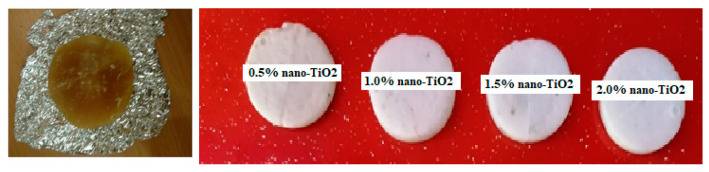
Samples of paraffin and nano-paraffins used in the study.

**Figure 4 nanomaterials-12-02266-f004:**
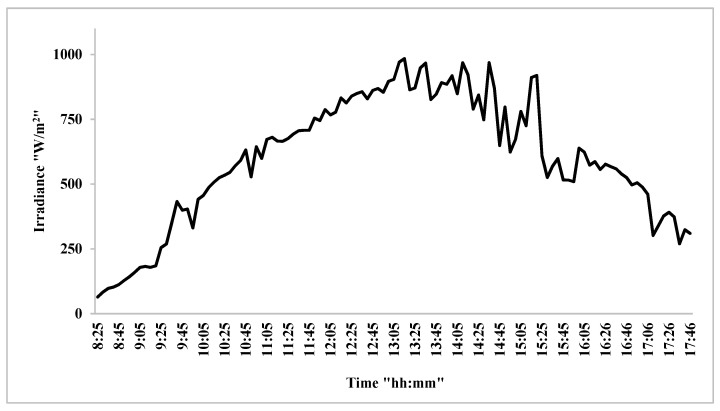
The variation of solar radiation intensity with time.

**Figure 5 nanomaterials-12-02266-f005:**
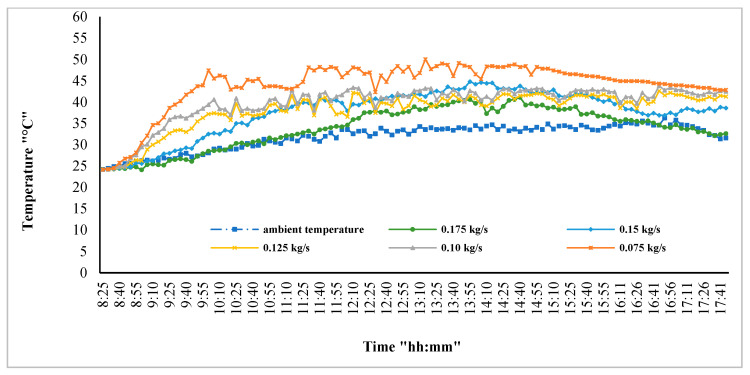
Water mass flow rate effect on PV module surface temperature.

**Figure 6 nanomaterials-12-02266-f006:**
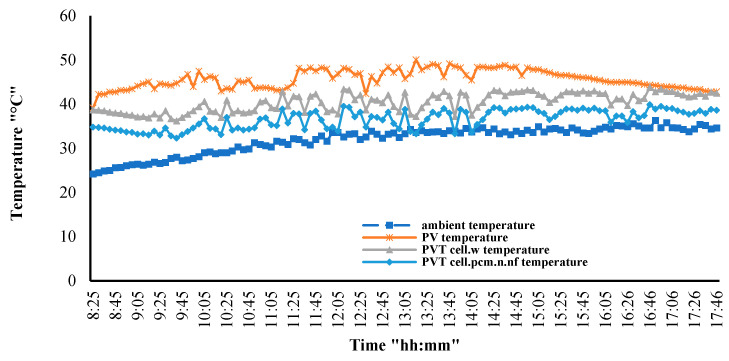
Cooling effect of the PV/T systems compared to the standalone PV module.

**Figure 7 nanomaterials-12-02266-f007:**
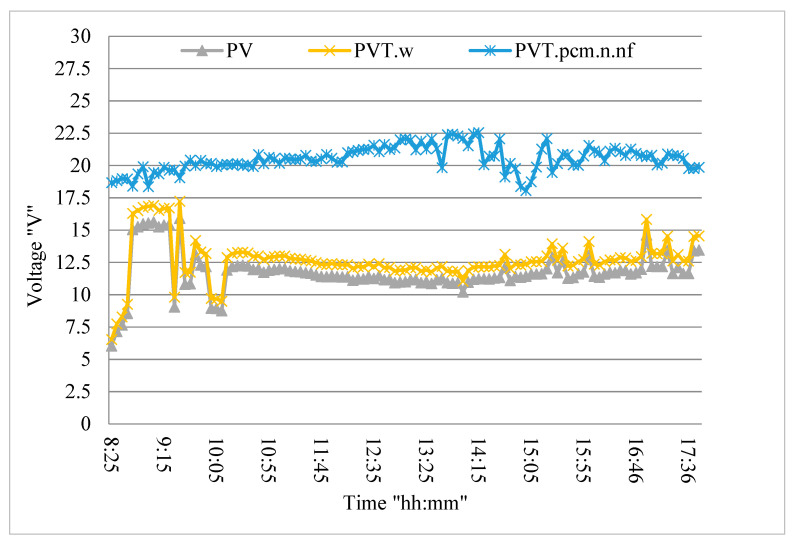
The tested systems’ voltage variation with time.

**Figure 8 nanomaterials-12-02266-f008:**
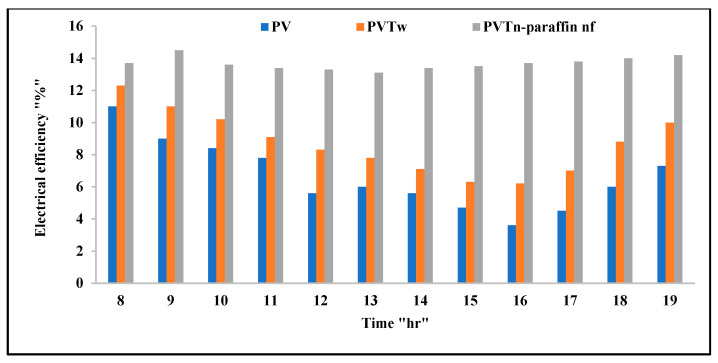
The electrical efficiencies of the tested systems variations with time.

**Figure 9 nanomaterials-12-02266-f009:**
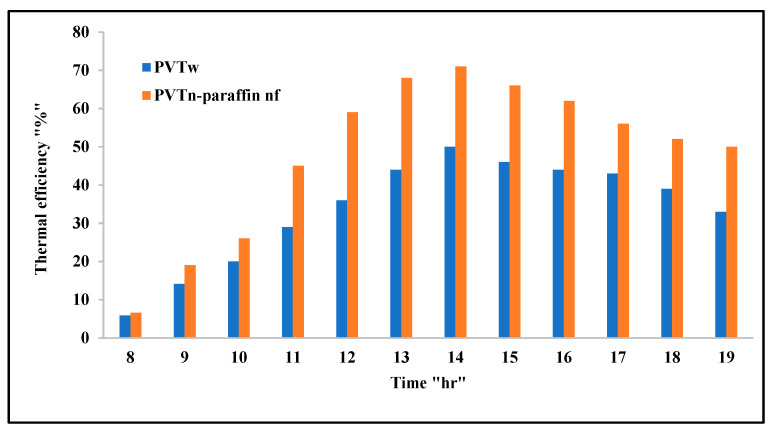
Variations of the thermal efficiencies of the tested systems with time.

**Figure 10 nanomaterials-12-02266-f010:**
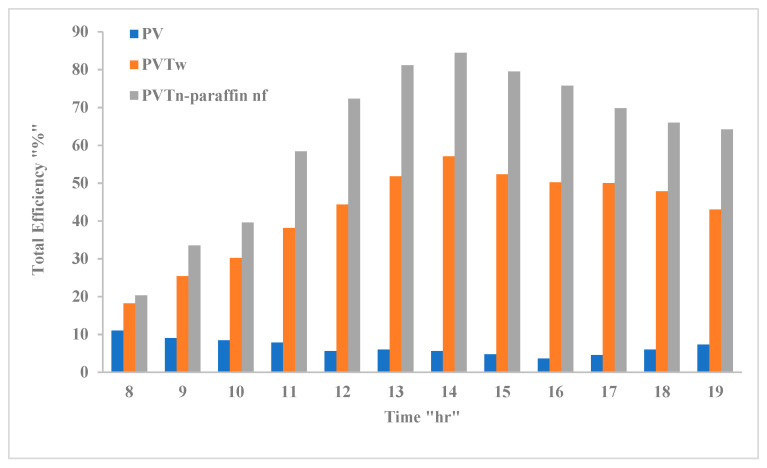
The total efficiencies of the tested systems variations with time.

**Figure 11 nanomaterials-12-02266-f011:**
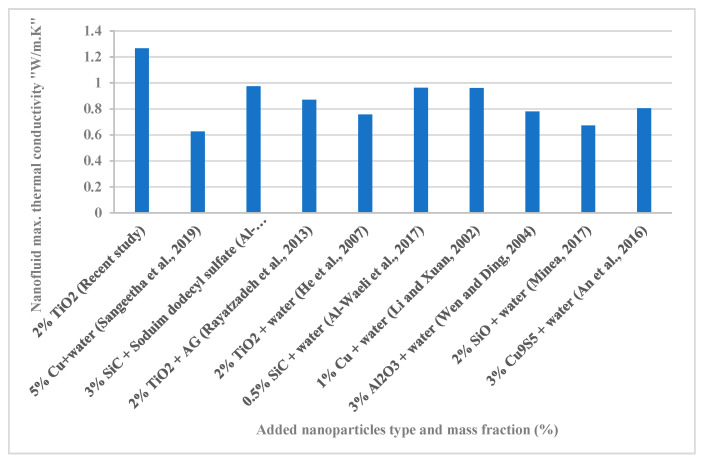
Comparison between the present study and other works in the literature, regarding nanofluid TC enhancement Sangeetha et al., 2019 [10]; Al-Waeli et al, 2019 [20]; Rayatzadeh et al., 2013 [32]; He et al., 2007 [49]; Al-Waeli et al., 2017 [57]; Li and Xuan, 2002 [60]; Wen and Ding, 2004 [61]; Minea, 2017 [62] and An et al., 2016 [63].

**Figure 12 nanomaterials-12-02266-f012:**
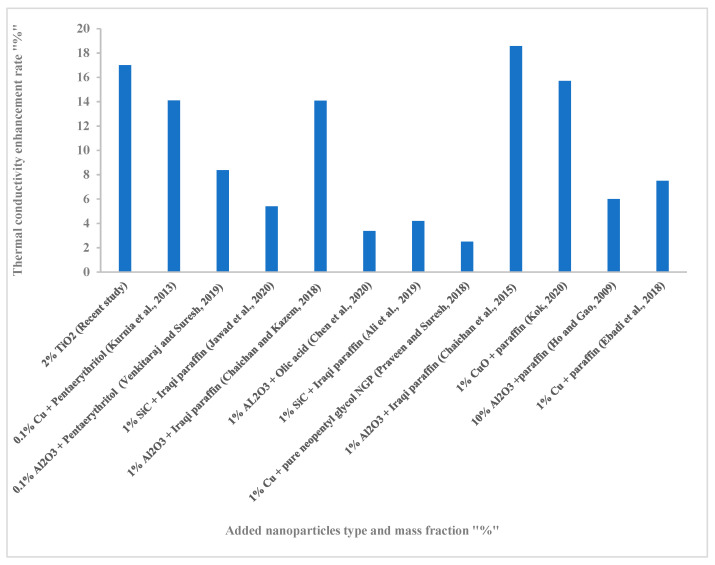
Comparison between the present study and other works in the literature, regarding nano-PCM TC enhancement Kurnia et al., 2013 [65]; Venkitaraj and Suresh, 2019 [66]; Jawad et al., 2020 [67]; Chaichan and Kazem, 2018 [68]; Chen et al., 2020 [44]; Ali et al., 2019 [69]; Praveen and Suresh, 2018 [70]; Chaichan et al., 2015 [64]; Kok, 2020 [71], Ho and Gao, 2009 [72] and Ebadi et al., 2018 [73].

**Table 1 nanomaterials-12-02266-t001:** Suggested solar energy stations with its details.

Governorate	Station Location	Designed Power (MW)	Voltage Level (KV)
Al-Muthana	Sawa-1	30	33
Al-Muthana	Sawa-2	50	132
Al-Muthana	Khidhir	50	132
Karbala	Karbala	300	132
Babilon	Al-Iskandaria	225	132
Diwania	Ramla	50	132
Wassit	Jassan	50	132

**Table 2 nanomaterials-12-02266-t002:** Specifications of the PV modules used in the study.

Electrical Properties (Standard Test Conditions)	Symbol	Specification
Model	-	STF–120P6
Max. power	Pmax	120 ± 3%
Voltage (Open-circuit)	Voc	22 V
Current (Short-circuit)	Isc	7.63 A
Voltage at maximum power	VMP	17.40 V
Current at maximum power	IMP	6.89 A
Max. electrical efficiency	$\eta_{ele.}$	14%

**Table 3 nanomaterials-12-02266-t003:** Nano-TiO_2_ specifications.

Item	Nano-TiO_2_
Manufacturer	Hongwu Nanometer
Appearance	White colored powder
Purity	99.7%
pH	7.4
Grain size (nm)	20–50 nm
Density (g/cm^3^)	3.92
Loss when dried %≤	0.23
Zeta potential (mV)	38.5
Molar mass (g/mole)	79.87
Melting point (°C)	1843
TC (W/m K)	280–750

**Table 4 nanomaterials-12-02266-t004:** Iraqi paraffin properties [56,57].

Property	Range	Units
Chemical composition	C20H42-C27H56	-
The temperature of melting point	43.5	°C
Liquid state density	836	kg/m^3^
Solid state density	929	kg/m^3^
Latent heat	197	kJ/kg
Solid state TC	0.21	W/m·K
Liquid state TC	0.19	W/m·K
Liquid state specific heat	2.15	kJ/kg K
Solid state specific heat	2.23	kJ/kg K

**Table 5 nanomaterials-12-02266-t005:** The details and uncertainties of the measuring devices used in the tests.

Measured Specification	Device	Uncertainty
Density	Density tester type (DII-300 L)	±0.16
Viscosity	Brookfield programmable viscometer (model: LVDV-III)	±0.12
TC	KD2 Pro-Analyzer	±0.8
Heat Capacity	KD2 Pro-Analyzer	±0.94
Temperatures	Thermocouples type K	±0.93
Weight	Delicate balance type (EJ6I0-E)	±0.34
Coolants flow rate	HC (US Hunter)	±0.55
Stability	Zeta-Sizer Nano Analyzer (ZSN)	±0.88

**Table 6 nanomaterials-12-02266-t006:** The thermophysical specification of the tested nanofluids at 25 °C.

Nanofluid Mass Fraction (%)	TC (W/m K)	Density (g/cm^3^)	Viscosity (mPa·s)	Stability Zeta Potential (mV)
Water	0.6	1.00	1.00	-
Nano-TiO_2_ (%) 0.5	0.82	1.0125	1.013	68
Nano-TiO_2_ (%) 1.0	1.13	1.019	1.023	64
Nano-TiO_2_ (%) 1.5	1.25	1.022	1.028	59
Nano-TiO_2_ (%) 2.0	1.36	1.025	1.033	48

**Table 7 nanomaterials-12-02266-t007:** The thermophysical properties of tested nano-paraffin at 65 °C.

Nanofluid Type Mass Fraction (%)	TC (W/m K)	Density (g/cm^3^)	Viscosity (mPa·s)	Stability TC Degradation (days)
Paraffin	0.2	950	0.088–0.077	-
Nano-TiO_2_ (%) 0.5	0.31	961.4	0.09–0.082	98
Nano-TiO_2_ (%) 1.0	0.48	967.86	0.92–0.085	94
Nano-TiO_2_ (%) 1.5	0.54	970.14	0.094–0.088	89
Nano-TiO_2_ (%) 2.0	0.66	972.61	0.0955–0.09	88

## Data Availability

Not applicable.

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
