# Peer review of "Adding Nano-TiO2 to Water and Paraffin to Enhance Total Efficiency of a Photovoltaic Thermal PV/T System Subjected to Harsh Weathers"

_nanomaterials, 2022, doi:10.3390/nano12132266_

Round 1

Reviewer 1 Report

In this article, the authors attempted to lower the temperature of photovoltaic modules by cooling them using two different methods, the first with the use of a coolant (water), and the second with a system consisting of a thermal tank containing nanoparaffin and cooled using a nanofluid. Then, based on the experiment, researchers showed an increase in the thermal conductivity of both water and paraffin, nano-TiO2. This is very important for the performance of a PV / T solar system, especially in difficult weather conditions. The work shows a clear improvement in its electrical efficiency by 106.5% and 57.7% more than the photovoltaic module (standalone) and the water-cooled photovoltaic system. This is a very important aspect, because the system presented in the work (nanoliquid and nanoparaffin) gives a greater possibility of controlling the thermal capacity and increasing both efficiencies (electrical and thermal) compared to a stand-alone photovoltaic module in difficult weather conditions.

The work was written in a very interesting way, a considerable study of the literature was made.

However, the minor remarks of the reviewer include

In fig. 1 fig a and fig b should be marked

In tables 3 and 4, superscript should be given in density

In table 4 in Solid state TC, the multiplication sign W / m · K is missing

In figure 3, 4, 5, 6, 7, 8 the units should be followed by a ";" semicolon

In figures 9 and 10, the units on the axles have to be completed.

Line 413 point 3.2 Comparison with other works from literature - there is a wrong entry, it should probably be 3.5

Line 432 point 5. Conclusions - there is a write error, should be 4

Author Response

Dear reviewer,

Thank you for useful comments and suggestions, all have been done so please see the revsed manuscript and the responce letter

Best regards

Reviewer 2 Report

Dear Authors,

the manuscript of article (nanomaterials-1570607 ) entitled "Adding nano TiO2 to water and paraffin to enhance total efficiency of a photovoltaic thermal PV/T system subjected to harsh weathers" by Miqdam T. Chaichan, Hussein A. Kazem, Ahmed A. Al-Amiery, Wan Nor Roslam Wan Isahak, Mohd S. Takriff has been reviewed. Manuscript can be published after major revisions. These specific comments should be addressed in the revised version:

  • what was the stability of the nanofluids produced in investigated temperature range, did it change with time?
  • Table 5, meaning of terms in the "Uncertainty" column should be explained,
  • what was the purpose of drying solid Al2O3 nanoparticles of 99.9% purity if it was later dispersed in water,
  • detailed schem of experimental setup should be provided including nests for temperature measurement,
  • how mas flow rate of nanofluid was determined? Could the Authors give average Reynolds number for flow inside investigated chanels,
  • what instrument was used to determine zeta potential?
  • what is origin of irradiance data presented in figure 3 ?
  • some typographic errors should be corrected,
  • definition of the  thermal and electrical efficiency should be provided,
  • the Authors should address issue of overal eficciency of system under investigation, including energy consumed for pumping of the nanofluid.

Your truly

Reviewer

Author Response

Dear reviewer,

Thank you for useful comments and suggestions, all have been done so please see the revsed manuscript and the response letter

Best regards

Reviewer 3 Report

The authors present a study of active cooling for PV panels for outdoor use in a hot climate. The work is of interest and is suitable for publication. However, I have some questions that the authors should address in the manuscript before publication:

The thermal conductivity enhancement of the paraffin by TiO2 addition is quite remarkable. How was this measured? The authors should provide details in the manuscript.

Some additional discussion of the design of the system should be presented especially as related to the thermal contact between the cooling material and the PV. Were there some kind of heat fins or other methods used to enhance the heat flow?

What does the "a" in Table 3 for the Crystal kind mean?

Can they comment on any effect expected for the durability of the pumps etc. from adding an oxide powder to the fluid?

Author Response

Dear reviewer,

Thank you for useful comments and suggestions, all have been done so please see the revised manuscript and the response letter

Best regards

Round 2

Reviewer 2 Report

Dear Authors,
revised version satisfy most requirements, so submitted manuscript can be accepted for publication in present form.

Yours truly
Reviewer 2